# EGR1/GADD45α Activation by ROS of Non-Thermal Plasma Mediates Cell Death in Thyroid Carcinoma

**DOI:** 10.3390/cancers13020351

**Published:** 2021-01-19

**Authors:** Seung-Nam Jung, Chan Oh, Jae Won Chang, Lihua Liu, Mi Ae Lim, Yan Li Jin, Yudan Piao, Hae Jong Kim, Ho-Ryun Won, Seong Eun Lee, Min Joung Lee, Jun Young Heo, Sangmi Jun, Doheon Lee, Woo Seok Kang, Dae-Woong Kim, Ki-Sang Rha, Young Il Kim, Yea Eun Kang, Bon Seok Koo

**Affiliations:** 1Department of Otolaryngology-Head and Neck Surgery, College of Medicine, Chungnam National University, Daejeon 35015, Korea; jungsn1122@cnu.ac.kr (S.-N.J.); strive1005@cnuh.co.kr (J.W.C.); dlaaldo22@cnu.ac.kr (M.A.L.); hryun83@cnuh.co.kr (H.-R.W.); ksrha@cnu.ac.kr (K.-S.R.); 2Department of Medical Science, College of Medicine, Chungnam National University, Daejeon 35015, Korea; ohchanny@o.cnu.ac.kr (C.O.); keaidian1992@cnu.ac.kr (L.L.); jinyanli5678@o.cnu.ac.kr (Y.L.J.); yudan0329@o.cnu.ac.kr (Y.P.); thename@cnu.ac.kr (H.J.K.); 3Research Center for Endocrine and Metabolic Diseases, School of Medicine, Chungnam National University Hospital, Chungnam National University, Daejeon 35015, Korea; seongeun316@cnu.ac.kr; 4Department of Biochemistry, Chungnam National University, Daejeon 34134, Korea; rmj1102@cnu.ac.kr (M.J.L.); jyheo@cnu.ac.kr (J.Y.H.); 5Convergent Research Center for Emerging Virus Infection, Korea Research Institute of Chemical Technology, Daejeon 34114, Korea; smjun@kbsi.re.kr; 6Department of Bio and Brain Engineering, KAIST, Daejeon 34141, Korea; dhlee@kaist.ac.kr; 7Department of Plasma Engineering Korea, Institute of Machinery & Materials, Daejeon 34103, Korea; kang@kimm.re.kr (W.S.K.); dwkim@kimm.re.kr (D.-W.K.); 8Department of Radiation Oncology, Chungnam National University Sejong Hospital, Sejong 30099, Korea; minesota@cnuh.co.kr; 9Department of Endocrinology and Metabolism, College of Medicine, Chungnam National University, Daejeon 35015, Korea

**Keywords:** NTPAM, thyroid cancer, ROS, EGR1, GADD45α

## Abstract

**Simple Summary:**

Recent studies have identified new anti-cancer mechanisms of nonthermal plasma (NTP) in several cancers. However, the molecular mechanisms underlying its therapeutic effect on thyroid cancer have not been elucidated. The objective of this study was to understand the anticancer effects of NTP-activated medium (NTPAM) on thyroid cancer cells and elucidate the signaling mechanisms responsible for NTPAM-induced thyroid cancer cell death.

**Abstract:**

(1) Background: Nonthermal plasma (NTP) induces cell death in various types of cancer cells, providing a promising alternative treatment strategy. Although recent studies have identified new mechanisms of NTP in several cancers, the molecular mechanisms underlying its therapeutic effect on thyroid cancer (THCA) have not been elucidated. (2) Methods: To investigate the mechanism of NTP-induced cell death, THCA cell lines were treated with NTP-activated medium -(NTPAM), and gene expression profiles were evaluated using RNA sequencing. (3) Results: NTPAM upregulated the gene expression of early growth response 1 (*EGR1*). NTPAM-induced THCA cell death was enhanced by EGR1 overexpression, whereas EGR1 small interfering RNA had the opposite effect. NTPAM-derived reactive oxygen species (ROS) affected EGR1 expression and apoptotic cell death in THCA. NTPAM also induced the gene expression of growth arrest and regulation of DNA damage-inducible 45α (*GADD45A*) gene, and EGR1 regulated *GADD45A* through direct binding to its promoter. In xenograft in vivo tumor models, NTPAM inhibited tumor progression of THCA by increasing EGR1 levels. (4) Conclusions: Our findings suggest that NTPAM induces apoptotic cell death in THCA through a novel mechanism by which NTPAM-induced ROS activates EGR1/GADD45α signaling. Furthermore, our data provide evidence that the regulation of the EGR1/GADD45α axis can be a novel strategy for the treatment of THCA.

## 1. Introduction

Thyroid cancer (THCA) is the most common type of endocrine cancer, and its incidence has been rapidly increasing over the last few decades [1,2]. THCA can be divided into five major types according to histopathological features: papillary, follicular, medullary, poorly differentiated, and anaplastic [3]. Papillary thyroid carcinoma (PTC) is the most common form of THCA, accounting for 80% of all thyroid malignancies [4]. PTC has a good prognosis and low mortality rate. However, PTC frequently metastasizes to the lymph node, which can consequently increase the loco-regional recurrence rate and cancer-specific mortality [5,6,7,8,9]. Anaplastic thyroid carcinoma (ATC), an undifferentiated thyroid carcinoma, exhibits the most aggressive clinical behavior because of its rapid growth, extensive local invasion of surrounding tissue, and metastasis to distant organs. Consequently, it has the poorest prognosis among thyroid carcinomas, with a median overall survival of only 2.5–6 months [10,11]. Therefore, it is necessary to discover novel ways to inhibit the aggressive features in THCA.

Nonthermal plasma (NTP) is an ionized gas composed of ions, electrons, free radicals, charged particles, and ultraviolet (UV) photons [12]. The anti-cancer effects of NTP have been reported in several in vitro and in vivo models and investigated as a novel anti-cancer therapeutic tool [13,14,15,16,17]. The most readily observed effect associated with NTP is oxidative stress, which is typically induced by the formation of reactive oxygen and nitrogen species (RONS) in cells [14,18]. Reactive oxygen species (ROS) generated by NTP can mediate the apoptosis of mammalian cancer cells by mitochondrial dysfunction [15].

Early growth response 1 (EGR1) belongs to the zinc finger transcription factor family and functions in cell growth, differentiation, and apoptosis. It is rapidly and transiently induced by a large number of growth factors, cytokines, ROS, UV, hypoxia, and ionizing radiation [19,20,21,22]. EGR1 exhibits either oncogenic or tumor-suppressive properties depending on the types of cells and stimuli [23,24,25]. EGR1 is implicated in the regulation of *TP53* gene expression and TP53 protein function in human melanoma cells leading to apoptosis [26,27,28], and EGR1 directly activates PTEN during irradiation-induced signaling [22]. EGR1 also directly regulates multiple tumor suppressors including *TGFB1*, *PTEN*, *TP53*, and *Fibronectin* [25]. In contrast, EGR1 transactivates the *Fibronectin* gene and enhances attachment of human glioblastoma cell line [29]. And high levels of EGR1 play a central role in the initiation of human prostate cancer [30,31].

The growth arrest and DNA damage-inducible gene 45 (*GADD45*) family consists of *GADD45A*, *GADD45B*, and *GADD45G*. GADD45 proteins are small (~18kDa) and highly acidic (pH 4.0–4.2) proteins and are mainly located within the nucleus. They are induced by genotoxic and certain cellular stress situations, including ionizing and UV radiation as well as various chemotherapeutic agents. GADD45 proteins have been implicated in a variety of growth regulatory mechanisms including cell cycle arrest, DNA repair, cell survival and senescence, and apoptosis. It was reported that GADD45α plays a role in negative regulation of cell cycle progression and cell growth in various cancer [24,32,33,34]. Related to thyroid cancer, there is one report that GADD45γ has potential to be used as a candidate gene for gene therapy in ATC for suppressing ATC cell growth [35].

Although recent studies have identified new mechanisms of NTP in several cancers, the molecular mechanisms underlying its therapeutic effects on THCA remain unclear. The objective of this study was to understand the inhibitory effects of NTP-activated medium (NTPAM) on THCA cell survival and elucidate the signaling mechanisms responsible for cell death.

## 2. Results

### 2.1. NTPAM Induces Apoptotic Cell Death in THCA Cells

The anti-cancer effects of NTP on cell growth, migration, and invasion of various cancers cells have been reported [15,36]. However, the molecular mechanisms underlying NTP-induced THCA cell death are still unclear. In this study, He and O_2_ gas mixture plasma was used, because He and O_2_ gas mixture plasma induces significantly more cell death than He gas plasma alone [13]. The technical specifications of the plasma system are shown in Figure 1A. Measurement of optical emission spectra in the NTP device with air gas flow was conducted at wavelengths of 280 to 920 nm by optical emission spectroscopy (Figure 1B). We measured temperature, electrical conductivity, pH, and RONS after NTP exposure. NTP treatment leads to the formation of hydrogen peroxide (H_2_O_2_), and nitric oxide (NO_3_^−^, NO_2_^−^) following continuous NTP exposure (Figure 1C). We conducted an experiment on the storage stability of NTPAM. NTPAM maintained its ability to suppress the viability of THCA cells for at least 10 days when stored at −80 °C and 2 days when stored at room temperature, suggesting that the anticancer effect of NTPAM could not be maintained for a long time (Appendix A). Therefore, we always used NTPAM freshly prepared in the same manner, prior to each experiment. First, we investigated whether NTPAM, a solution produced by exposing a liquid medium to NTP, could induce anti-cancer effects in various THCA cells. We examined the viability of BCPAP, HTh7, KTC2, 8505C, and FRO-Luc cells after NTPAM treatment for 24 h. NTPAM effectively inhibited the viability of all THCA cells in an activation time-dependent manner (namely 0–120 min) (Figure 1D). Figure 1E shows the cell morphology changes including marked rounding, shrinkage, and detachment from the culture dish after NTPAM treatment of BCPAP and 8505C cells. Several studies have reported that NTP induces apoptotic cancer cell death [16,36]. To determine whether NTPAM led to the induction of apoptosis, AnnexinV/PI staining was performed at 24 h after 2 h dose-NTPAM treatment in THCA cells. NTPAM treatment resulted in a significant increase in apoptosis compared with the control group (Figure 1F). In addition, NTPAM treatment triggered caspase 3 activation and PARP cleavage, indicating the induction of apoptosis (Figure 1G). These results suggest that NTPAM induced apoptotic cell death in the THCA cell lines.

### 2.2. NTPAM Induces Mitochondrial Dysfunction in THCA Cells

Our previous study demonstrated that THCA cell lines have reduced mitochondrial function compared with the Nthy-ori3-1 normal cell line [37]. Thus, we hypothesized that the effects of NTPAM treatment on mitochondrial function of cancer cells may differ from those on normal thyroid cells. First, we investigated the morphological changes after NTPAM treatment of Nthy-ori3-1 and BCPAP cells. BCPAP cells exhibited elevated accumulation of swollen, abnormal mitochondria with disrupted cristae compared with Nthy-ori3-1 cells after NTPAM treatment. After NTPAM treatment, the percentage of dysmorphic mitochondria in BCPAP cells increased significantly compared to Nthy-ori3-1 cells (Figure 2A). These results showed that NTPAM induced mitochondrial stress, with this effect stronger in THCA cells than in normal thyroid cells. Additionally, we compared the oxygen consumption rates (OCRs) between Nthy-ori3-1, BCPAP, and 8505C cells after NTPAM treatment. There was no significant change in basal respiration OCR, maximal respiration OCR, or ATP-dependent respiration OCR in Nthy-ori3-1 cells after NTPAM treatment. However, THCA cell lines exhibited reduced mitochondrial function compared with Nthy-ori3-1 cells (Figure 2B–E). These results suggest that the greater effect of NTPAM in cancer cells may be related to decreased mitochondrial function in cancer cells compared to normal cells at baseline.

### 2.3. EGR1 Is a Regulator of Thyroid Tumorigenesis

To determine the major mechanism of NTPAM in human THCA, we conducted transcriptome analysis of THCA cells with or without 2 h dose-NTPAM treatment for 24 h. As shown in Figure 3A,B, changes in BCPAP and 8505C cells were positively correlated with NTPAM treatment compared to other cells. There were many significantly changed genes after NTPAM treatment in BCPAP and 8505C cells (Appendix A). Our in vitro data revealed abnormal mitochondrial accumulation in THCA after NTPAM treatment. Therefore, we identified an association between NTPAM treatment and the mitochondrial unfolded protein response (UPRmt) in human THCA. Interestingly, NTPAM treatment induced significant changes in genes related to UPRmt (Figure 3C). To explore the important relationships between NTPAM and biological processes, we evaluated significantly enriched KEGG gene sets with statistical significance (Figure 3D). As shown in Figure 3E and Appendix A, the top 10 upregulated pathways and top 10 downregulated pathways in BCPAP and 8505C cells revealed upregulation of mitophagy and ribosomal biogenesis as well as downregulation of pyruvate metabolism in BCPAP cells and oxidative phosphorylation in 8505C cells after NTPAM treatment. To identify the upstream regulators after NTPAM treatment in BCPAP and 8505C cells, we conducted ingenuity pathway analysis (IPA). Interestingly, EGR1, DNA damage-inducible transcript 3 (DDIT3), and activating transcription factor 4 (ATF4) were identified as transcription regulators in both BCPAP cells and 8505C cells after NTPAM treatment, and EGR1 was selected as a gene for further study (Figure 3F), because it showed the highest expression ratio and z score in BCPAP and 8505C cells equally. We also investigated two other candidate genes, ATF4 and DDIT3 (Appendix A). First, we investigated the expression of gene sets regulated by EGR1. As shown in Figure 3G and Appendix A, NTPAM induced significant differences in genes related to EGR1 expression. To assess the usefulness of EGR1 expression related with phenotypes in PTC, we investigated the expression of EGR1 in a THCA cohort (Figure 3H). We determined a high EGR1 group and a low EGR1 group based on the median value of EGR1 in the TCGA-THCA cohort. The expression of EGR1 was significantly higher in normal samples compared to tumor samples. To evaluate the relationship between EGR1 expression and clinicopathological features, we catalogued the clinicopathological features associated with samples from 500 samples in the THCA cohort. The chi-square test and Fisher’s exact test were used to evaluate the significance of the correlation of EGR1 expression with clinical and pathological parameters. Low EGR1 expression was significantly associated with several clinicopathological parameters including male gender (*p* = 0.034), tumor size (*p* = 0.014), and lymph node metastasis (*p* = 0.009) (Appendix A). Additionally, we performed the multivariable regression analysis between the high EGR1 expression and clinicopathologic factors using TCGA-THCA cohort (Appendix A). In multivariable analysis, lymph node metastasis was the significant factor associated with EGR1 expression. Next, the Kaplan-Meier analysis were performed from total 500 patients. EGR1 low group showed slightly unfavorable prognosis, however, there was no significant differences between two group (Figure 3I).

### 2.4. EGR1 Is Activated by NTPAM to Mediate Cell Death in THCA

EGR1 can function as a tumor suppressor or an oncogene, depending on the type of tumor cell [23,24,25]. However, the role of EGR1 in THCA remains poorly understood. To validate the RNA-seq analysis results (Figure 4A), the mRNA and protein expression levels of EGR1 were examined in BCPAP and 8505C cells after 2 h dose-NTPAM treatment for 24 h. EGR1 expression levels were markedly increased in NTPAM-treated cells (Figure 4B,C). We hypothesized that EGR1 could suppress tumor progression in the thyroid. To this end, the protein expression levels of EGR1 were investigated in normal and tumor tissues derived from the same THCA patients. Lower expression of EGR1 was detected in four tumor samples relative to normal tissue (Figure 4D). Then, we explored the effects of transient overexpression and knockdown of EGR1 on the cell viability of THCA cells. An EGR1 overexpression vector was transiently transfected into BCPAP and 8505C cells, and cell viability was determined by cell counting. EGR1 overexpression considerably suppressed cell viability compared with the control (Figure 4E). In contrast, to examine the effect of EGR1 knockdown, cells were transiently transfected with EGR1-specific siRNA or negative control siRNA and EGR1 knockdown caused an increase in cell viability (Figure 4F). These results confirm that EGR1 suppresses the cell viability of THCA cells and is activated by NTPAM.

### 2.5. EGR1 Promotes NTPAM-Mediated Cell Death through Regulation of ROS

NTP can generate several RONS such as NO_3_^−^, hydrogen, O_2_, and ozone and NTP treatment mediates an increase in cellular ROS, contributing to cell death in various cancers [15,17,38,39,40]. In this study, we also explored whether NTPAM mediates ROS accumulation in THCA cells. Mitochondrial ROS levels increased in 2 h dose-NTPAM-treated cells and N-acetylcysteine (NAC), a general anti-oxidant chemical, partly or completely reversed the effects of NTPAM treatment (Figure 5A). Corresponding to the change in ROS levels, the cell viability of BCPAP and 8505C cells were reduced by NTPAM treatment and rescued by cotreatment with NAC (Figure 5B). NTPAM treatment-mediated ROS increased EGR1 expression, and NAC inhibited NTPAM-induced EGR1 induction (Figure 5C). EGR1 expression also increased in a dose-dependent manner upon NTPAM treatment (Figure 5D). NTPAM-induced THCA cell death was enhanced by EGR1 overexpression, whereas EGR1 siRNA led to the opposite result (Figure 5E,F). These data indicate that the upregulation of EGR1 by NTPAM induces ROS-mediated cell death in THCA.

### 2.6. NTPAM Induces Cell Death through the ROS/EGR1/GADD45α Pathway in THCA

Recent studies have reported that EGR1 transcription factor directly regulates the expression of tumor suppressors including TGFβ1, PTEN, p53, and fibronectin [22,25,26,27,28,41]. Also, EGR1 directly binds and regulates GADD45α and GADD45β expression during UVB stress or oxidative stress such as arsenic exposure [21,24]. We used RNA-seq to confirm the target genes of EGR1 upregulated by NTPAM. GADD45α, but not PTEN, TP53, or TGFβ1, was reproducibly and statistically significantly upregulated in BCPAP and 8505C cells (Figure 6A). The mRNA and protein levels of GADD45α were increased after 2 h dose-NTPAM treatment for 24 h, as shown by RNA-seq (Figure 6B,C). Transfection with EGR1-specific siRNA led to the downregulation of GADD45α, whereas EGR1 overexpression caused an increase in GADD45α expression (Figure 6D,E). To determine if the induction of GADD45α is mediated by the direct binding of EGR1 to the promoter of the GADD45α, we evaluated the effect of EGR1 on the activity of the human GADD45α promoter. EGR1 siRNA decreased the luciferase activity of the wild-type GADD45α promoter construct, whereas EGR1 overexpression increased the activity. However, EGR1 did not affect mutant GADD45α promoter activity (Figure 6F,G). Thus, direct DNA binding of EGR1 to the GADD45α promoter appears to be important for transactivation of the GADD45α promoter. We investigated the role of GADD45α in the 8505C cell line. GADD45α overexpression considerably suppressed cell viability compared with the control. In contrast to GADD45α overexpression, GADD45α knockdown caused an increase in cell viability (Figure 6H,I). These results suggest that NTPAM treatment-mediated ROS upregulates EGR1, and EGR1 increases the expression of GADD45α through transcriptional regulation.

### 2.7. Antitumor Effect of NTPAM in THCA in a Xenograft Mouse Model

To confirm the effects of NTPAM treatment on THCA cells in an established xenograft mouse model (Reg. No. CNUH IRB No. 202006A-CNU-112), bioluminescence imaging was performed to observe changes in tumor cell growth in vivo. We subcutaneously administered Luciferase-expressing FRO (FRO-Luc) cells into BALB/c-nude mice and treated them with NTPAM for 3 weeks. The region of interest (ROI) was measured on day 21 post-injection (Figure 7A). The ROI of the NTPAM-treated group was significantly reduced compared to the control group (Figure 7B,C). After final treatment with NTPAM, tumor weight was significantly reduced in the NTPAM-treated group compared with the control media injection group (Figure 7D,E). To determine whether NTPAM induced THCA cell death through the upregulation of EGR1/GADD45α in vivo, we analyzed changes at the molecular level in EGR1 and GADD45α by immunohistochemistry. EGR1 and GADD45α were increased in NTPAM-treated mouse tumor tissues compared with control tissues (Figure 7F). This is consistent with the in vitro protein expression data after NTPAM treatment. These results suggest that NTPAM exerts an inhibitory effect against tumor growth in vivo model. Additionally, we confirmed that NTPAM did not cause any side effects on normal tissue in a mouse model. NTPAM-related damage was not found in several organs including the liver, heart, lung, spleen, and kidney in treated mice (Appendix A). Our all results indicate that NTPAM more effectively inhibited THCA cell growth than did the control, and that ROS mediated the NTPAM-induced apoptotic cell death through the regulation of EGR1/GADD45α (Figure 7G).

## 3. Discussion

Recent studies have shown that NTP can induce cell death in various cancers such as those of the brain, colon, skin, lung, pancreatic, and head and neck [14,16,38,42]. This study investigated the anticancer effects and mechanism of NTP, a potential next-generation therapeutic tool for THCA.

NTP, which has been used for cancer treatment, has been applied using direct and indirect techniques. In the direct method, the cells and mouse tumor are directly exposed to NTP. In the indirect method, the cells are exposed to cell culture medium treated with NTP. Recent studies have shown that although the effectiveness of direct plasma treatment is greater than that of indirect treatment, NTPAM (NTP-activated medium) shows great potential for tumors located inside the body, suggesting that it may be an alternative to direct plasma exposure in cancer treatment [43,44]. Therefore, in this study, we used NTPAM to investigate the effect of NTP on THCA.

A promising cancer therapy should not only be effective but also selective. NTP selectively induces cancer cell death but not normal cell death [45,46]. There is a hypothesis to explain the selective killing of cancer cells by NTP. RONS created by NTP can trigger mechanisms that lead to apoptosis. ROS are generally produced in cancer cells at much higher levels than in normal cells. Thus, cancer cells are more sensitive than normal cells to ROS generated by NTP [18]. In our study, the concentrations of hydrogen peroxide and nitric oxide in media peaked in the measurable range after only 1 h of NTP exposure. And NTPAM pH reaches ~8.6 after 2 h of activation with NTP. We should consider that our results may in part be due to the effects of cell exposure to supraphysiological pH, though our findings focus on ROS after NTPAM treatment. NTPAM treatment-mediated oxidative stress, resulting in mitochondrial ROS accumulation following apoptotic cell death in THCA. We also determined whether NTPAM destroys mitochondria of cancer cells without damaging normal cells. BCPAP cells exhibited elevated accumulation of swollen, abnormal mitochondria with disrupted cristae compared with Nthy-ori3-1 cells after NTPAM treatment. NTPAM induced mitochondrial stress, and this effect was stronger in THCA cells than in normal thyroid cells. There was no significant change in basal respiration OCR, maximal respiration OCR, or ATP-dependent respiration OCR in Nthy-ori3-1 cells after NTPAM treatment. However, THCA cell lines exhibited reduced mitochondrial function compared with Nthy-ori3-1 cells. Additionally, we confirmed that NTPAM did not cause any side effects on normal tissue in a mouse model. NTPAM-related damage was not found in several organs including the liver, heart, lung, spleen, and kidney in treated mice (Appendix A).

To determine the major mechanism of NTPAM in human THCA, we conducted transcriptome analysis of THCA cells with or without NTPAM treatment. Transcriptome analysis revealed that there were many significantly changed genes after NTPAM treatment in THCA cells. Interestingly, EGR1, DDIT3, and ATF4 were identified as upstream transcription regulators through IPA, and EGR1 was selected as a gene for this study because it showed the highest expression ratio and z score in BCPAP and 8505C cells equally. However, we also investigated two other candidate genes, ATF4 and DDIT3. The expression of ATF4 and DDIT3 increases in response to microenvironmental stresses including oxidative stress and endoplasmic reticulum stress, and ATF4 induces the transcriptional expression of DDIT3. ATF4 controls the expression of adaptive genes that allow cells to endure stress. However, ATF4 promotes the induction of apoptosis under persistently stressful conditions [47]. We found that ATF4 siRNA reduced cancer cell viability and NTPAM-induced cell death was enhanced by ATF4 siRNA, suggesting that ATF4 may be increased by NTPAM for THCA cell adaptation but not cell death (Appendix A).

EGR1 is a redox-sensitive transcription factor, and its mRNA and protein expression is rapidly induced by ROS [48,49]. In this study, EGR1 was markedly upregulated by NTPAM in both BCPAP and 8505C cells. NTPAM treatment-mediated ROS increased EGR1 gene expression and NAC, a ROS scavenger, reduced NTPAM-induced EGR1 induction as well as NTPAM-induced cell death. EGR1 overexpression significantly suppressed cell viability compared with the control, indicating that EGR1 acts as a tumor suppressor in THCA cells. NTPAM-induced THCA cell death was enhanced by EGR1 overexpression, whereas EGR1 siRNA had the opposite effect. However, the change of cell viability by NTPAM showed even in EGR1 siRNA, suggesting that NTPAM-induced cancer cell death may be the result of simultaneous regulation of various molecules and EGR1 is at least one of the major factors for NTPAM-induced cell death.

We found that the expression of EGR1 was significantly higher in normal samples than in tumor samples in the THCA cohort. We also investigated the correlation between EGR1 expression and clinicopathologic characteristics. Low EGR1 expression in PTC is significantly associated with several clinicopathological parameters including male gender, tumor size, and lymph node metastasis, suggesting the novel role of EGR1 as a tumor suppressor in PTC. In addition, we showed that EGR1 expression was related to the disease-free survival rate in PTC through the analysis using a THCA cohort.

GADD45 is a regulator at the G2/M checkpoint and plays roles in apoptosis, the DNA damage response, and cell-cycle arrest. It is often upregulated in response to various environmental stresses and drug therapies [50,51]. It was reported that the increase of GADD45α in response to As^3+^ was mediated sequentially by ELK1 and EGR1 [21]. Also, EGR1 directly binds to GADD45α and GADD45β and regulates their expression during UVB stress or oxidative stress [21,24]. In our data, GADD45α was significantly upregulated by NTPAM in BCPAP and 8505C cells, as shown by RNA-seq. EGR1 positively regulated GADD45α expression via direct binding to the GADD45α promoter. GADD45α overexpression suppressed cell viability compared with the control, whereas GADD45α knockdown caused an increase in cell viability, suggesting that GADD45α acts as a tumor suppressor. Our results clearly demonstrate that NTPAM treatment-mediated ROS upregulates EGR1 and EGR1 increases the expression of GADD45α through transcriptional regulation, leading to cell death in THCA.

Heterogeneity of tumor genotypes among patients and within the same patient may render target therapeutics ineffective [52]. In this study, we examined the effect of NTPAM on different types of thyroid cancer cell lines with a different genetic variation. Even though papillary THCA (PTC) and anaplastic THCA (ATC) are totally different disease clinically and the cellular mechanism underlying might be different between these cell line, NTPAM consistently suppressed the growth of both PTC (BCPAP) and ATC (8505C, HTh7, and KTC2), characterized by the different types of genetic alterations. In addition, in a xenograft in vivo model using FRO-Luc cells, an anaplastic THCA, we also found that tumor growth in the NTPAM-treated group was significantly reduced compared to the control.

The stability of NTPAM for anticancer effect during its storage is an important factor for determining its potential for clinical applications. We conducted an experiment on the storage stability of NTPAM. NTPAM maintained its ability to suppress the viability of THCA cells for at least 10 days when stored at −80 °C and 2 days when stored at room temperature, suggesting that the anticancer effect of NTPAM could not be maintained for a long time (Appendix A). Therefore, research to maintain NTPAM activity for a long time is urgently needed to overcome the limitations for clinical application.

NTP has some advantages over conventional chemotherapy and radiotherapy. NTP as a therapeutic agent can be generated safely and inexpensively. NTPAM is suitable for the clinical applications because NTPAM can be prepared in advance and stored until use [53]. In addition, NTPAM could be a useful treatment for large area, such as an intraperitoneal treatment to eliminate the residual micro-tumors and the micro-metastatic dissemination of cancer cells after surgery [43]. Although NTPAM treatment shows novel and effective tools for cancer therapy, further investigations are still needed to more fully determine the clinical anti-tumor effects of NTPAM.

This is the first study to elucidate transcriptomic changes in NTPAM-treated THCA cells. Our findings establish that EGR1 as a potential tumor suppressor that leads to NTPAM-induced apoptotic cell death via GADD45α regulation in THCA cells. We conclude that the application of NTPAM may be an effective and safe treatment for THCA in the future.

## 4. Materials and Methods

### 4.1. Cell Lines and Materials

The human THCA cell lines BCPAP, HTh7, KTC2, 8505C, and FRO-Luc and the normal thyroid cell line Nthy-ori3-1 were obtained from the Young Joo Park Laboratory (Seoul National University, Seoul, Korea). All THCA cells were authenticated using the short tandem repeat typing method in May 2020. BCPAP and HTh-7 cells were maintained in high-glucose Dulbecco’s modified Eagle medium (Gibco, Gaithersburg, MD, USA). KTC2, 8505C, FRO-Luc, and Nthy-ori3-1 cells were maintained in Roswell Park Memorial Institute 1640 medium (Gibco). All cells were supplemented with 10% fetal bovine serum and 100 units/mL penicillin-streptomycin (Gibco) and grown at 37 °C with 5% carbon dioxide under humidified conditions.

### 4.2. Experimental System Specifications of NTP and NTPAM Preparation

Each medium was treated using an NTP jet generated by a device composed of a quartz tube (outer diameter, 6 mm; inner diameter, 4 mm) with two electrodes—an inner stainless-steel tube electrode and an outer ground ring-electrode. The inner tube electrode also functions as a gas inlet. A high voltage amplifier supplied input power (a few kV at 20 kHz) to the device. Discharge power varied from 10 to 24 W. The plasma jet was positioned perpendicularly over the medium in a flask, and the gas flowed from the inside of the device towards the medium surface. Helium (He) (4 standard liters per minute (slpm)) and oxygen (O2) (1 standard cubic centimeters per minute (sccm)) were used as carrier gases, because several reports previously found that the addition of O2 improved the efficiency of cancer cell inhibition [13,54]. The emission spectra of the plasma revealed the presence of strong nitrogen-, He-, and O2-related light emissions. To prepare NTPAM, RPMI 1640 medium was exposed to NTP for various periods of time which correspond to different NTPAM concentrations. Most of the expreriments were conducted using 2 h dose-NTPAM. 2 h dose-NTPAM represents the NTPAM produced by exposing to NTP for 2 h. The distance between the plasma device and above the medium was maintained at around 1 cm. NTPAM was prepared with 50 mL of culture media. Cell treatment with NTPAM was performed by changing the culture medium for NTPAM. We always used NTPAM freshly prepared in the same manner, prior to each experiment.

### 4.3. Cell Viability Assay

Cells were seeded at a density of 5 × 10^3^ cells per well in triplicates in 96-well plates. Next day, the medium was replaced, and cells were treated with 2 h dose-NTPAM for 24 h. After treatment, cell viability was measured using WST-1 (Roche Diagnostics, Indianapolis, IN, USA) according to the manufacturer’s protocol. WST-1 formazan was quantitated at 450 nm using an enzyme-linked immunosorbent assay reader. Results are presented as percentages relative to control cells.

### 4.4. Apoptosis Assay

To detect apoptosis, Annexin V-fluorescein isothiocyanate (FITC) and propidium iodide (PI) staining was performed, and cells were evaluated using a flow cytometer (LSRFortessa X-20; BD Biosciences, Brea, CA, USA) with an Alexa FluorTM 488 Annexin V/Dead Cell Apoptosis kit (Invitrogen, Carlsbad, CA, USA) following the manufacturer’s protocol. In short, cells were treated with NTPAM and incubated for 24 h, after which they were collected, washed with Dulbecco’s phosphate-buffered saline, stained with Annexin V-FITC and PI, and analyzed using flow cytometry. FlowJo software version 10 (Tree Star, Ashland, OR, USA) was used to analyze the percentage of cells in four populations: FITC-/PI- (living cells), FITC+/PI- (early apoptotic cells), FITC+/PI+ (late apoptotic cells), and FITC-/PI+ (necrotic cells).

### 4.5. Measurement of Oxygen Consumption Rate

Oxygen consumption rate (OCR) was measured using a Seahorse XF-24 analyzer (Seahorse Bioscience Inc., North Billerica, MA, USA). Briefly, thyroid cells were seeded in 5 replicates in XF-24 plates (4 × 10^4^ cells in 200 μL of growth medium per well) and placed in a 37 °C/5% CO_2_ incubator for 24 h. The sensor cartridge was placed into calibration buffer supplied by Seahorse Bioscience and incubated at 37 °C in a non-CO_2_ incubator for 24 h before the experiment. Immediately before measurement, the cells were washed and incubated in assay medium at 37 °C for 1 h in a non-CO_2_ incubator. Oligomycin A (20 µg/mL), an ATP synthase (complex V) inhibitor, was injected to measure cellular ATP production and carbonyl cyanide m-chloro phenyl hydrazine (optimized concentration, 50 μM). The uncoupling agent was used to measure maximal respiration by disrupting the mitochondrial membrane potential. Rotenone, a complex I inhibitor, was injected to inhibit mitochondrial respiration (20 μM), which allowed calculation of non-mitochondrial respiration. The OCR was automatically recorded by a sensor cartridge and calculated using Seahorse XF-24 software.

### 4.6. Transmission Electron Microscopy

Cells were pelleted via centrifugation, and the cell pellet was fixed in 2.5% glutaraldehyde in 0.1 M phosphate buffer (pH 7.4) at 4 °C. After 2 h, the pellet was rinsed three times in 0.1 M phosphate buffer and then post-fixed in 1% osmium tetroxide at 4 °C for 1–2 h, followed by three washes in phosphate buffer. The pellet was dehydrated via a series of ethanol solutions and exposed to two changes of propylene oxide. After the dehydration process, the pellet was infiltrated with 2:1, 1:1, and 1:2 mixtures of EMbed 812 resin and propylene oxide for 1 h each time and then finally embedded in 100% EMbed 812 resin. After polymerization for 24–48 h at 60 °C, ultrathin plastic sections (80-nm thick) were cut at room temperature using a Leica EM UC6 ultramicrotome (Leica Microsystems GmbH, Wetzlar, Germany) and collected on 200-mesh carbon-coated grids. The grids were post-stained with 2% uranyl acetate and 1% lead citrate at room temperature for 15 and 5 min, respectively. The Zeiss LEO912AB 120 kV transmission electron microscope (Carl Zeiss, Oberkochen, Germany) and FEI Tecnai G2 Spirit Twin 120 kV transmission electron microscope (FEI Company, Columbia, MD, USA) were used for transmission electron microscopy observations.

### 4.7. RNA Extraction for Sequencing

To analyze the transcriptome and identify differentially expressed genes (DEGs), RNA was extracted from THCA cells. After 2 h dose-NTPAM treatment for 24 h, THCA cells or control cells were isolated and total RNA was extracted using an easy-spin Total RNA Extraction kit (iNtRON, Gyeonggi, Korea) following the manufacturer’s protocol. All experiments were conducted under clean conditions, and the equipment was pre-autoclaved. Sequencing was performed by Macrogen, Inc (Seoul, Korea). Brifely, The extracted RNA quality was evaluated using an Agilent 2100 Bioanalyzer RNA Nano Chip (Agilent Technologies, Savage, MD, USA). A total of 1 μg of extracted RNA was used to construct RNA libraries with a TruSeq Stranded mRNA Sample Preparation Kit v2 (Illumina, San Diego, CA, USA) according to the manufacturer’s protocols. The library quality was analyzed with the Agilent 2100 Bioanalyzer using an Agilent DNA 1000 kit. All samples were sequenced on the Illumina HiSeq 2500 platform (Illumina), yielding an average of 38 million paired-end 100 base-pair reads.

### 4.8. Bioinformatic Transcriptome Analysis

All genomic data of PTC from The Cancer Genome Atlas were obtained from a specific portal (https://tcga-data.nci.nih.gov) and cancer browser (https://genome-cancer.ucsc.edu). Gene expression data from mRNA and clinical parameters in patients with PTC (n = 500) were analyzed. The reads were aligned to the UCSC Homo sapiens reference genome (GRCh37/hg19) using TopHat2 v2.1.5 (https://ccb.jhu.edu/software/tophat/index.shtml). The default TopHat parameter options were used. To analyze the DEG profiles between the compared groups (control vs. NTPAM treatment), the Tuxedo protocol was used [55]. The aligned reads were processed via Cufflinks v2.2.1 (https://github.com/cole-trapnell-lab/cufflinks), which is based on the fragments per kilobase of exon model per million reads mapped (FPKM), and unbiased, normalized RNA sequencing (RNA-seq) fragment counts were used to analyze the relative transcript levels [55]. Gene transfer format (GTF) files were generated to quantitatively compare the transcript levels in each sample to those in a reference GTF file. Next, we used Cuffdiff to calculate the differences in FPKM between each group pair. False discovery rate-adjusted P-values were calculated via the Benjamini-Hochberg multiple testing method [56]. To compare gene sets in relation to NTPAM, the gene set collection of the Kyoto Encyclopedia of Genes and Genomes (KEGG) was obtained from Enrichr (https://amp.pharm.mssm.edu/Enrichr/). DEGs and KEGG pathways with corrected *p*-values < 0.05 according to the Benjamini-Hochberg procedure were considered statistically significant. Heat maps were generated using PermutMatrix version 1.9.3 (http://www.lirmm.fr/~caraux/PermutMatrix/).

### 4.9. Western Blot Analysis

Cells were lysed in Ripa buffer containing 150 mM sodium chloride, 1.0% Nonidet P-40, 0.5% sodium deoxycholate, 0.1% sodium dodecyl sulfate, 50 mM Tris (pH 8.0), and a protease inhibitor cocktail (Roche Applied Science, Vienna, Austria). Electrophoresis was performed as previously described [57]. The following primary antibodies were used for Western blot analysis: anti-poly (ADP-ribose) polymerase (PARP), anti-caspase 3, anti-B-cell lymphoma 2 (Bcl-2), anti-EGR1, anti-β-actin (1:1000; Cell Signaling Technology, Danvers, MA, USA), and anti-GADD45α (1:1000; Santa Cruz Biotechnology, Santa Cruz, CA, USA). Following incubation with the corresponding horseradish peroxidase-conjugated secondary antibodies (1:1000; Santa Cruz Biotechnology), immunoreactive bands were visualized using enhanced chemiluminescence (Bio-Rad Laboratories, Hercules, CA, USA). The original Western Blot pictures can be found in the Appendix A.

### 4.10. RNA Isolation and Quantitative PCR

Total cellular RNA was extracted using TRIzol reagent (Invitrogen). The cDNAs were synthesized with 5 μg total RNA and TOPscriptTMRT DryMIX (Enzynomics Inc., Dajeon, Korea) according to the manufacturer’s instructions. Amplification was carried out using SYBR Green qPCR Master Mix (Thermo Fisher Scientific). The PCR reactions were performed for 40 cycles of 95 °C for 15 s, 60 °C for 1 min, and 72 °C for 1 min. melting curve analysis was performed. The PCR primer sequences were as follows: human EGR1-F: 5′-ATG ATC CCC GAC TAC CTG TTT-3′, EGR1-R: 5′-CTG AGT GGC AAA GGC CTT AAT-3′ (amplicon length: 144 bp); human GADD45α-F: 5′-GAG AGC AGA AGA CCG AAA G-3′, GADD45α-R: 5′-AGA GCC ACA TCT CTG TCG T-3′ (amplicon length: 186bp); and human GAPDH-F: 5′-GTC TCC TCT GAC TTC AAC AGC G-3′, GAPDH-R: 5′-ACC ACC CTG TTG CTG TAG CCA A-3′ (amplicon length: 131bp). We used 2^−ΔΔCt^ model for Relative quantification of real-time qPCR fold change. The Ct values provided from real-time PCR instrumentation are imported into Microsoft Excel. Here are the formulae for 2^−ΔΔCt^ method. ΔCt_ctrl_ = (Ct_ctrl_ − Ct _GAPDH_), ΔCt_NTPAM_ = (Ct_NTPAM_ − Ct_GAPDH_). ΔΔCt_ctrl_ = ΔCt_ctrl_ − ΔCt_ctrl_ average, ΔΔCt_NTPAM_ = ΔCt_NTPAM_ − ΔCt_ctrl average_. 2^−ΔΔCtctrl^, 2^−ΔΔCtNTPAM^ is relative value. Fold change = > 2^−ΔΔCtctrl^/2^−ΔΔCtctrl^, 2^−ΔΔCtNTPAM^/2^−ΔΔCtctrl^ [58].

### 4.11. Patients’ Samples and Ethics Statement

Four patients’ smaples who were diagnosed with PTC in Chungnam National University Hospital were included in the study. Paired tumor and normal thyroid tissues were obtained from the patients with PTC. All specimens were collected from patients after informed consent was obtained in accordance with the institutional guidelines of Chungnam National University Hospital. The protocol for this study was approved by the Institutional Review Board of Chungnam National University Hospital (Reg. No. CNUH IRB No. 2017-07-005). Human tissues were homogenized in RIPA buffer and processed according to the western blot analysis protocol.

### 4.12. Measurement of Mitochondrial ROS Production

The MitoSOX fluorescent probe (Cat. #M36008; Thermo Fisher Scientific, Waltham, MA, USA) was used to detect mitochondrial superoxide. Cells were co-treated with or without NTPAM and N-acetyl cysteine (NAC) (2 and 5 mM) and then incubated at 37 °C for 24 h. After treatment, cells were stained with MitoSOX (5 μM) in Hank’s balanced salt solution for 15 m. Fluorescence-stained cells (1 × 10^4^) were analyzed via flow cytometry (LSRFortessa X-20; BD Biosciences), and FlowJo software version 10 (Tree Star) was used to analyze the percentage of cells.

### 4.13. Transient Transfection

Cells were seeded at 2 × 10^5^/well in 6 well and cultured overnight to achieve 60–70% confluence. Next day, Transient transfection was performed using Lipofectamine RNAiMAX reagent (Invitrogen) with small interfering RNA (siRNA) and jetPEI DNA transfection reagent (Polyplus, Illkirch-Graffenstaden, France) for the overexpression vector, following the manufacturers’ standard protocols. The siRNAs for EGR1 and GADD45α were acquired from Santa Cruz Biotechnology, and the siRNA for the control was acquired from Bioneer (Daejeon, Korea). The overexpression vectors for EGR1 and GADD45α were obtained from OriGene Technologies (Rockville, MD, USA). GADD45 WT-luc and GADD45 P-2/3m-luc were obtained from Addgene (Watertown, MA, USA) [38].

### 4.14. Xenograft In Vivo Tumor Model

Mice were kept under specific pathogen-free conditions and used in accordance with the guidelines of the Institutional Animal Care and Use Committee of Chungnam National University, which approved the animal research protocol (Daejeon). Ten BALB/c-nude mice (6 weeks old) were obtained from Orient Bio Inc. (Seongnam, Korea). Luciferase-expressing FRO (FRO-Luc) cells were inoculated subcutaneously into the lower right flanks of BALB/c-nude mice. The mice were divided randomly into control medium and NTPAM groups (5 mice per group) 10 days later when the tumors reached ~50 mm in diameter, and the tumors were injected daily with 100 μL of control medium or NTPAM for 21 days. After the last injection of NTPAM, the tumors were excised from euthanized mice and used for western blot assays.

### 4.15. Bioluminescence Imaging and Survival Analysis

Thyroid tumor growth was observed via bioluminescence imaging using an in vivo imaging system (IVIS) consisting of a Lumina XRMS instrument (PerkinElmer, Waltham, MA, USA). For the detection of in vivo live imaging, the mice were intraperitoneally administered 150 mg/kg D-luciferin (Promega, Fitchburg, WI, USA). After anesthetizing the mice with 2% isoflurane in 100% O_2_, bioluminescence images were taken. The signals were analyzed and quantified by calculating the luminescence intensity in the region of interest (ROI) using Living Image software (Caliper Life Sciences, Waltham, MA, USA) registered with the imaging system.

### 4.16. Histological and Immunohistochemical Analysis

Tissue samples were fixed in 4% formalin solution and paraffin-embedded. For H&E staining, tissue sections were deparaffinized in xylene, hydrated in graded alcohols and stained with hematoxylin and eosin. The samples examined under an automatic digital slide scanner (Pannoramic MIDI) after mounting. For immunohistochemistry (IHC), tissue sections were deparaffinized in xylene, hydrated in graded alcohols, and heated (100 °C) for 15 min in Antigen Retrieval Citra Soluion, pH 6.0 for antigen retrieval. For single immunostaining, endogenous peroxidase activity was blocked in a 1% hydrogen peroxide solution (Sigma-Aldrich, St. Louis, MO, USA) in PBS with 0.3% Triton X-100 for 30 min at room temperature. The sections were incubated with the indicated antibodies overnight at 4 °C and then incubated with the corresponding horseradish peroxidase-conjugated secondary antibody. Finally, 3,3′ diaminobenzidine (DAB; DAKO) was used to detect these labeled antibodies and the nucleus was stained with hematoxylin. After rinsing with PBS, the samples were mounted and analyzed by an automatic digital slide scanner (Pannoramic MIDI).

### 4.17. Statistical Analysis

Data were analyzed using GraphPad Prism 8.0 software (GraphPad Software, La Jolla, CA, USA) and SPSS statistical softmare for Windows, version 22 (International Business Machines, Armonk, NY, USA). Non-parametric one-way ANOVA followed by Dunnett’s post-hot test, Mann-Whitney U test, Student’s *t*-test, Independent *t*-test, and logrank test were used for statistical analysis. Data were expressed as mean ± standard deviation of the mean (±SD). Differences were considered relevant at *p* < 0.05 (* *p* < 0.05, ** *p* < 0.01, *** *p* < 0.001).

## 5. Conclusions

In conclusion, NTPAM treatment showed beneficial anticancer effects through upregulation of the EGR1/GADD45α signaling in THCA. Our findings demonstrate that EGR1 is a potential tumor suppressor that leads to apoptotic cell death in THCA and the EGR1/GADD45α axis could be a good therapeutic target for THCA.

## Figures and Tables

**Figure 1 cancers-13-00351-f001:**
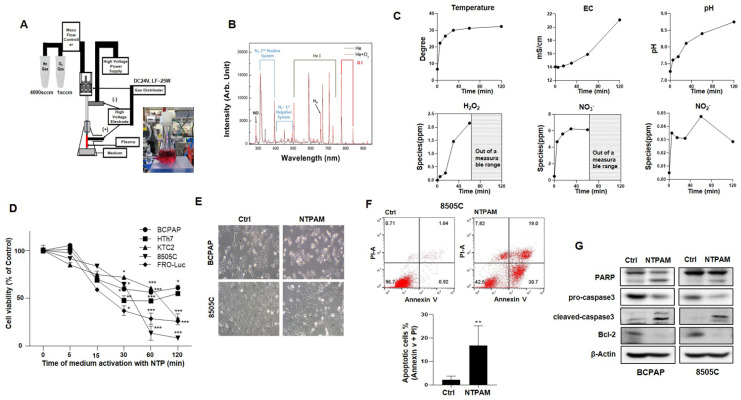
Apoptotic effects of nonthermal plasma-activated medium (NTPAM) on thyroid cancer (THCA) cells. (**A**) Diagram of the developed He- and O_2_-based NTP system and scheme as well as a photograph depicting the experimental design and preparation of NTPAM. (**B**) Measurement of the optical emission spectra (OES) of the NTP device with air gas flow. (**C**) Medium temperature, electrical conductivity (EC), pH, and concentrations of NO_3_^−^, NO_2_^−^, and H_2_O_2_ following continuous NTP exposure. (**D**) Various THCA cells were treated with indicated concentration (0, 5, 15, 30, 60, 120 min of activation) of NTPAM for 24 h, and cell viability was analyzed using the WST1 assay. Results were analyzed using non-parametric one-way ANOVA followed by Dunnett’s post-hoc test. (**E**) Cell morphology changes in BCPAP and 8505C cells after 2 h dose-NTPAM treatment for 6 h. (**F**) Apoptosis was determined by AnnexinV/PI flow cytometry in 8505C cells after 2 h dose-NTPAM treatment for 24 h. Results were analyzed using Mann-Whitney *U* test. (**G**) Apoptotic proteins were examined using western blot analysis after 2 h dose-NTPAM treatment for 24 h. Each figure is representative of three independent experiments. Data were expressed as mean ± standard deviation of the mean (±SD). Differences were considered relevant at *p* < 0.05 (* *p* < 0.05, ** *p* < 0.01, *** *p* < 0.001).

**Figure 2 cancers-13-00351-f002:**
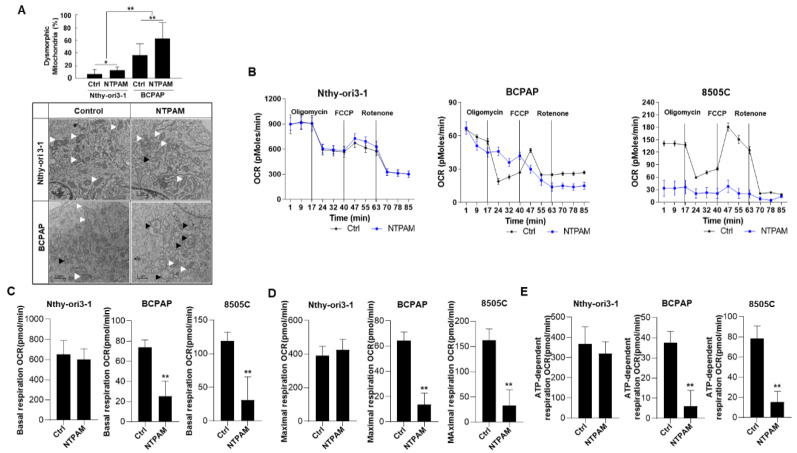
NTPAM induces mitochondrial dysfunction in THCA cells. (**A**) Electron microscopy examination of Nthy-ori3-1 and BCPAP cells to evaluate dysmorphic mitochondria with or without 2 h dose-NTPAM treatment for 1 h. Normal mitochondria: white arrow, dysmorphic mitochondria: black arrow. Results were analyzed using Mann-Whitney U test. (**B**) Oxygen consumption rates (OCRs) measured in Nthy-ori3-1, BCPAP, and 8505C cells with or without 2 h dose-NTPAM treatment for 1 h. (**C**) Basal respiration, (**D**) Maximal respiration, (**E**) ATP-dependent respiration in Nthy-ori3-1, BCPAP, and 8505C cells with or without NTPAM treatment. Results were analyzed using Mann-Whitney *U* test. Results were analyzed using Mann-Whitney *U* test. Each figure is representative of three independent experiments. Data were expressed as mean ± standard deviation of the mean (±SD). Differences were considered relevant at *p* < 0.05 (* *p* < 0.05, ** *p* < 0.01).

**Figure 3 cancers-13-00351-f003:**
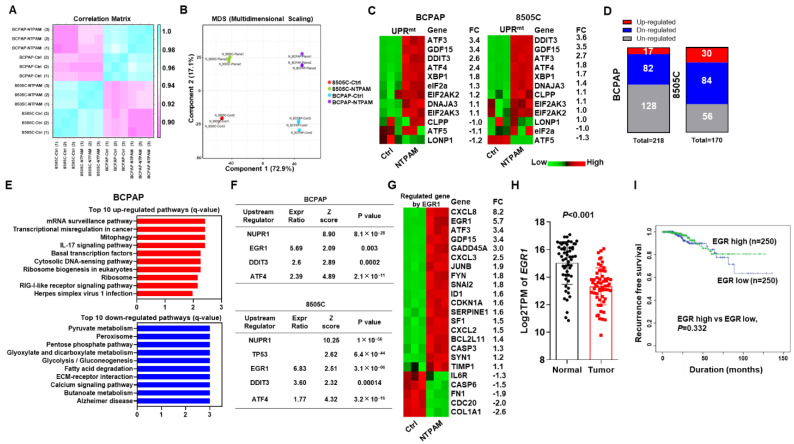
Transcriptome analysis of THCA cells after NTPAM treatment identifies EGR1 as a regulator of thyroid tumorigenesis. (**A**) Correlation matrix of BCPAP and 8505C cells with or without NTPAM treatment. (**B**) Multi-dimensional scaling plots for BCPAP and 8505C cells with or without NTPAM treatment. (**C**) Heatmap analysis of significantly changed genes related to the UPRmt in BCPAP and 8505C cells with *p* < 0.05 for comparisons of cells with or without NTPAM treatment. (**D**) The total number of significantly enriched KEGG gene sets with corrected *p*-values < 0.05 according to the Benjamini-Hochberg procedure (considered statistically significant). (**E**) Top 10 upregulated pathways and top 10 downregulated pathways in BCPAP cells in relation to NTPAM treatment. (**F**) Upstream regulators in relation to NTPAM treatment in BCPAP and 8505C cells via IPA. (**G**) Heatmap analysis of significantly changed genes related to EGR1 with *p* < 0.05 compared to BCPAP cells with or without NTPAM treatment. (**H**) Box plot of log2TPM of EGR1 values between normal and tumor samples in the TCHA cohort. The significance for between group differences was evaluated using logrnak test. (**I**) Disease-free survival in relation to EGR1 expression in the THCA cohort. TPM: transcripts per million.

**Figure 4 cancers-13-00351-f004:**
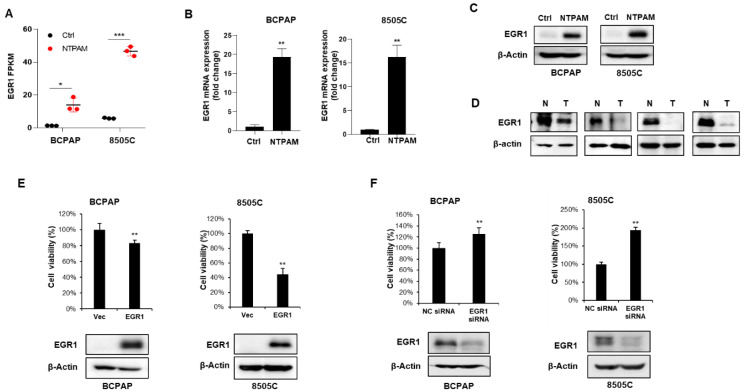
NTPAM-induced EGR1 inhibits THCA cell viability. (**A**) Changes in EGR1 expression after NTPAM treatment, as determined by RNA-seq. Results were analyzed using independent *t*-test. After 2 h dose-NPTAM treatment for 24 h, the EGR1 expression was examined by quantitative PCR (qPCR) analysis (**B**) and western blot analysis (**C**). Results were analyzed using Mann-Whitney U test. (**D**) Tissue samples obtained from patients with THCA were examined using western blot analysis with an anti-EGR1 antibody. (**E**) BCPAP and 8505C cells were transiently transfected with an EGR1 overexpression vector or a negative control vector for 48 h. Cell viability was analyzed by cell-counting analysis. Results were analyzed using Mann-Whitney U test. (**F**) BCPAP and 8505C cells were transiently transfected with EGR1-specific siRNA or negative control siRNA for 48 h. Cell viability was analyzed by cell-counting analysis. Results were analyzed using Mann-Whitney U test. Each figure is representative of three independent experiments. Data were expressed as mean ± standard deviation of the mean (±SD). Differences were considered relevant at *p* < 0.05 (* *p* < 0.05, ** *p* < 0.01, *** *p* < 0.001).

**Figure 5 cancers-13-00351-f005:**
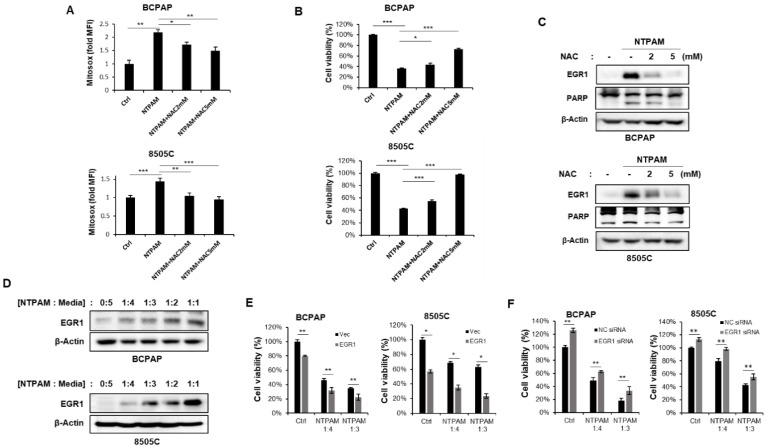
NTPAM induces cell death through the regulation of ROS-induced EGR1 in THCA cells. BCPAP and 8505C cells were treated with 2 h dose-NTPAM for 6 h or cotreated with NTPAM and NAC. (**A**) Mitochondrial ROS levels were measured by Mitosox-based flow cytometry. Results were analyzed using non-parametric one-way ANOVA followed by Dunnett’s post-hoc test. (**B**) Cell viability was evaluated using the WST1 assay. Results were analyzed using non-parametric one-way ANOVA followed by Dunnett’s post-hoc test. (**C**) Cell lysates were assessed by western blot analysis using antibodies against EGR1, PARP, and β-actin. (**D**) BCPAP and 8505C cells were treated with NTPAM for 24 h in a dose-dependent manner up to 1:4 dilution and cell lysates were assessed by western blot analysis using an antibody against EGR1. (**E**) BCPAP and 8505C cells were transfected with or without EGR1 vector for 48 h, and then cells were treated with NTPAM for 24 h at the indicated concentrations. Cell viability was analyzed using the WST1 assay. Results were analyzed using Mann-Whitney *U* test. (**F**) BCPAP and 8505C cells were transfected with or without EGR1-specific siRNA for 48 h, and then cells were treated with NTPAM for 24 h at the indicated concentrations. Cell viability was analyzed using the WST1 assay. Results were analyzed using Mann-Whitney U test. Each figure is representative of three independent experiments. Data were expressed as mean ± standard deviation of the mean (±SD). Differences were considered relevant at *p* < 0.05 (* *p* < 0.05, ** *p* < 0.01, *** *p* < 0.001).

**Figure 6 cancers-13-00351-f006:**
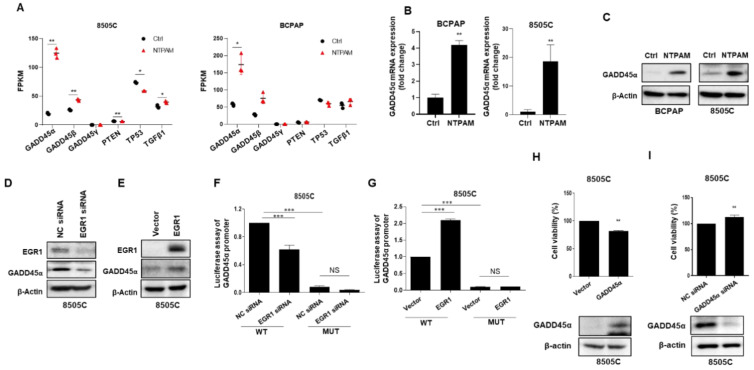
NTPAM induces cell death through the ROS/EGR1/GADD45α pathway in THCA cells. (**A**) Changes in the expression of EGR1 target genes after NTPAM treatment, as determined by RNA-seq. Results were analyzed using independent *t*-test. After treatment of BCPAP and 8505C cells with 2 h dose-NTPAM for 24 h, GADD45α was examined using qPCR analysis (**B**) and western blot analysis (**C**). Results were analyzed using Mann-Whitney *U* test. Cells were transfected with EGR1 siRNA (**D**) or the EGR1 overexpression vector (**E**) for 48 h, followed by western blot analysis. EGR1 siRNA (**F**) or the EGR1 overexpression vector (**G**) was co-transfected with the wild-type GADD45α promoter or the mutant GADD45α promoter P-2/3m into 8505C cells. At 24 h after transfection, cells were lysed, and luciferase activity was measured using a luminometer (FilterMax F5). Results were analyzed non-parametric one-way ANOVA followed by Dunnett’s post-hoc test. 8505C cells were transiently transfected with a GADD45α overexpression vector (**H**) or the GADD45α-specific siRNA (**I**) for 48 h. Cell viability was evaluated using the WST1 assay. Results were analyzed using Mann-Whitney U test. Each figure is representative of three independent experiments. Data were expressed as mean ± standard deviation of the mean (±SD). Differences were considered relevant at *p* < 0.05 (* *p* < 0.05, ** *p* < 0.01, *** *p* < 0.001).

**Figure 7 cancers-13-00351-f007:**
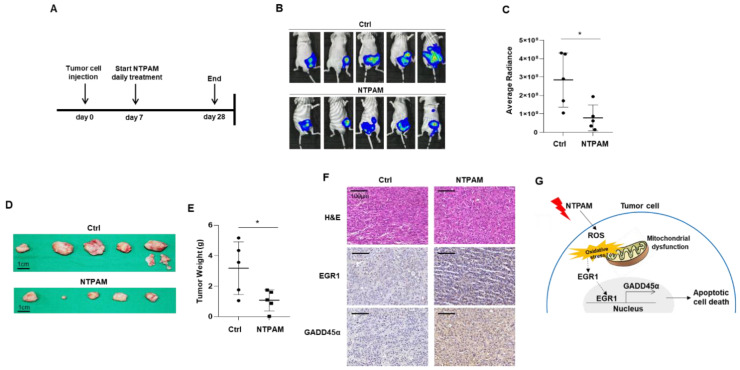
NTPAM exerts anticancer effects in a xenograft in vivo model. Luciferase-expressing FRO (FRO-Luc) THCA cells were injected subcutaneously into BALB/c-nude mice and treated with NTPAM every day for 3 weeks. (**A**) Schematic illustration of the in vivo experimental design. (**B**) Final tumor images of cancer cells tracked with the IVIS imaging system following the injection of mice with FRO-Luc cells. (**C**) The luminescence radiance of the ROI of the tumor was determined. Results were analyzed using Mann-Whitney U test. (**D**) Final tumor images. (**E**) Final tumor weight. Tumor tissues were extracted after the final determination of tumor growth. Results were analyzed using Mann-Whitney *U* test. (**F**) Representative images of H&E and immunohistochemical staining of control and NTPAM group. (**G**) The proposed NTPAM-induced ROS/EGR1/GADD45α axis in THCA based on this study. Data were expressed as mean ± standard deviation of the mean (±SD). Differences were considered relevant at *p* < 0.05 (* *p* < 0.05).

## Data Availability

We didn’t produce any public data base. There isn’t public data during this study

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
