# Peer review of "EGR1/GADD45α Activation by ROS of Non-Thermal Plasma Mediates Cell Death in Thyroid Carcinoma"

_cancers, 2021, doi:10.3390/cancers13020351_

Round 1
Reviewer 1 Report
In my opinion overall the article is quite solid and deserves publication.
The authors must address the following points before submitting the revised version:
1- Figure 1G: there is a series of Western blots to indicate that NTAPM produces apoptosis. I suggest looking for activated Caspase 3, which is expected to increase. I think this could be more conclusive than just showing the decrease in pro-Caspase 3.
2- Section 2.2 starts with:
"Previously, we demonstrated that THCA cell lines have reduced mitochondrial function compared with the Nthy-ori3-1 normal cell line (unpublished data). Thus, we hypothesized that the effects of NTPAM treatment on mitochondrial function of cancer cells may differ from those on normal thyroid cells".
This data (unpublished) should be included, in the main text or as Supplementary material. The hypothesis of the whole section is based in observations that are not shown anywhere.
3- Section 2.5: adding the plot of mitochondrial ROS (Mitosox) in the NTPAM + NAC condition is needed to corroborate that the antioxidant effect of NAC can actually be seen with Mitosox. 4- Information about qPCRs is required to evaluate the experimental results. The authors do not mention how they calculate the fold change from the Ct, nor delta delta Ct or anything similar. If it can be of help I suggest reading the MIQE guidelines: https://academic.oup.com/clinchem/article/55/4/611/5631762Author Response
Please see the attachment.

Reviewer 2 Report
The authors reported on the molecular mechanisms underlying its therapeutic effect on thyroid cancer using cell lines. Several papers have published concerning the molecular mechanism on this topic. The authors are trying to demonstrate the molecular changes using various thyroid cancer cell types.
A few suggestion and questions:
1) Clinically, PTC and ATC have totally different prognosis. They are totally different disease clinically. Subsequently, the cellular reaction and molecular mechanism underlying might be different between these cell lines. Please clearly discuss this point.
2) Define the abbreviations. Please avoid using abbreviations in the abstract without defining ("ROS").
Author Response
Reviewer 2:
The authors reported on the molecular mechanisms underlying its therapeutic effect on thyroid cancer using cell lines. Several papers have published concerning the molecular mechanism on this topic. The authors are trying to demonstrate the molecular changes using various thyroid cancer cell types.
A few suggestion and questions:
1. Clinically, PTC and ATC have totally different prognosis. They are totally different disease clinically. Subsequently, the cellular mechanism underlying might be different between these cell lines. Please clearly direction and molecular discuss this point.
: Response
- Thank you for valuable comments and we agree with the reviewer’s point. As you recommended, we added the description about your comments in Discussion section (12page, Line342-346).
2. Define the abbreviations. Please avoid using abbreviations in the abstract without defining ("ROS").
: Response
-Thank you for your comments. We have defined an abbreviation for ROS (reactive oxygen species) in Abstract section (2page, Line48).
We reviewed the abbreviation in all sentences.
Reviewer 3 Report
This is an in-depth study of NTPAM effects on thyroid cancer cells in vitro and in vivo. Both phenomenological and mechanistic investigations were undertaken to identify the ROS/EGR1/GADD45α axis as one of the key mechanisms of cell-killing effect of NTPAM, which is proposed to be a potential therapeutic target for aggressive thyroid cancers. The study is well-designed and performed on a high methodological and technical level. The manuscript presents abundant data from very distinct technologies, looks solid and logically balanced.
Before recommending the paper for publication, shortcomings need to be eliminated as follows.
General comments
- First, the manuscript title does not sound smooth (e.g. “cell death of thyroid carcinoma” is awkward), and second, the title implies that NTPAM is the source of RONS (indeed, L111-112 and Fig. 1B demonstrate this) rather than RONS inducer. However, in many places in the manuscript, NTPAM is namely referred to as the RONS (or ROS) inducer (e.g. L203, 235, 306, 330, etc.). Since the experiments aimed at distinguishing the role of NTPAM as of a ROS supplier or inducer (or both) were not performed (or remained beyond the scope of the study), it is necessary to be consistent and correct; for example, the authors may wish to use “NTPAM treatment-mediated oxidative stress” terminology or similar.
- Since many potential readers may be not familiar with NTP and NTPAM, it is necessary to provide more evidence that the method of NTPAM preparation reproducibly yields the medium of similar quality, characteristics and properties that do not vary significantly between different batches. The authors mention something in this regard on L342-344 of the Discussion but “data not shown”. This kind of information should be essentially added to supplementary material or a reference/references should be provided if available.
- Figure legends are too succinct. It is necessary to add more information, first of all regarding the time of cell exposure to NTPAM and type of statistical tests applied whenever applicable. Otherwise it is too difficult to follow the content of the experimental procedure and the way of the result evaluation/interpretation.
- In the Methods, please explain whether after cell incubation with NTPAM, the medium was changed to the complete one (see e.g. L120, difficult to understand) or there have been other culture manipulations. In the text, please change wording from “dose-dependent” to “time-dependent” when different exposure time intervals are meant (e.g. L116).
- NTPAM pH reaches ~8.6 after 2h of activation with NTP. While the work focuses on oxidative stress after NTPAM treatment, were the effects of cell exposure to supraphysiological pH ruled out? This may be added to supplementary material and discussed.
- Many results may and need to be added to supplementary material.
Specific comments
- L39 does not flow well, please rephrase.
- L52: should be “direct binding to its promoter”.
- L55: please add “evidence” (or an equivalent term) after “provide”.
- L62: “poorly differentiated” is also a recognized histological type of thyroid carcinoma that should be added; note also that these are types, not subtypes.
- L89 does not flow well, please rephrase.
- Fig.1A: “media” à “medium”. Was the medium stirred during NTP treatment? In Fig.1A legend, “NTP” à “NTPAM”.
- Fig.1D: again, concentration is not measured in minutes; it is difficult to understand whether the medium was changed after NTPAM treatment and then the cells were kept for 24h in a normal medium; what statistical test was used? Please write clearly.
- Fig.2A: potential readers may not be familiar with the appearance of normal and dysmorphic mitochondria, so please indicate those by e.g. arrowheads of different color. How long was NTPAM treatment, what was total incubation time, what statistical test was used?
- Fig.2C-E: what statistical tests was applied?
- L163-166: here, in addition to EGR1, the authors mention the DDIT and ATF4 transcripts as upstream regulators of significantly altered pathways (inferred from transcriptome analysis as shown in Fig.3F), and state that EGR1 was chosen for further analysis (why? because of the highest fold-change? – please clarify). Along with that, in the Discussion (L296-303), DDIT and ATF4 are mentioned again in the “data not shown” mode. It is necessary to place the findings on DDIT and ATF4 to supplementary material. If the authors do so, then the DDIT and ATF4-related part in the Discussion can be safely removed or discussed just briefly.
- L168-178: evaluation of EGR1 relationship to clinicopathological features of TCGA thyroid cancer cohort. The definitions of “low” and “high” EGR1 expression is not given (likely, the median was chosen as a cutoff value as follows from Supplementary table 1), so please add this. Also please calculate odds ratios (binary features) and a hazard ratio (DFS) and p-values using multivariable regression models (adjusted at least for sex and age) for EGR1. This would provide more evidence for the associations.
- Fig.3I. Is this a comparison of pooled data on EGR1 from RNA-seq of normal and PTC tissues from TCGA? Please clarify this, specify the statistical test and add the comparison of matched (i.e., from the same patient) normal thyroid and cancerous tissues.
- Fig.3I, the Kaplan-Meier analysis. Why the number of cases for low- and high-expression cases is 128 each (are these the upper and the lower quartiles?)? The manuscript mentions a total of 500 cases (L172) - please clarify. Also, there is no need to show confidence limits (dotted lines?) on the graph or otherwise please explain what these lines are in the figure legend.
- Fig.4A-B. Please specify the statistical test(s).
- Fig.4B: qPCR data. My quick check of EGR1 primers shows that the amplicon length is >600 bp (please correct me if I am wrong; Sequence ID: NM_001964.3, EGR1-F 607-626+/+ and EGR1-R 1222-1203 +/-, thus 1222-607=615bp). Such size is not well suitable for qPCR (usually expected to be from 70 to 200bp) and therefore it is difficult to judge how reliable the result is (for GADD45α the amplicon length is 186 bp based on NM_001924.4, thus acceptable). It is strongly recommended to re-measure relative EGR1mRNA expression by qPCR using shorter amplicon(s) (if possible) or to provide compelling data on the assay validity in supplementary material.
- L189-190 and Fig.4D. Four tumor samples from patients are mentioned. The patients are not described in the Methods (please make sure that ethical regulations were met!). The method of protein extraction from human tissues is not described. Please, add those.
- L197: data in this paragraph do not address cell death due to EGR1 overexpression, please remove “to mediate cell death in THCA”.
- L203: as mentioned in #1, it is unclear whether NTPAM induced ROS production or supplied those. Please “production” to another more neutral term, e.g. “accumulation”.
- Fig5ABEF: please specify the statistical tests.
- Fig5A-F (all): how long was NTPAM treatment?
- L222: recommend adding “reproducibly and statistically” before “significantly”.
- Fig.6ABFGHI: please indicate the length of exposure to NTPAM (also in CDE) and specify the statistical tests.
- The 2.7 subsection from L242 (xenograft model). FRO-luc cells were used for these experiments. So far, the effect of NTPAM exposure on these cells were not described in the manuscript or supplementary material (or I could not find). It is absolutely necessary to provide evidence of the existence of at least of the basic effects in FRO-luc cells in vitro, similar to those observed in 8505 and BCPAP.
- L247: please expand ROI.
- Related to the Fig.7F: were there any signs of cell death on H&E-stained tissue sections?
- Fig.7CE: please specify statistical tests.
- Fig.7D: please add a scale (may be in cm).
- L265: please add “potential” before “next generation”, otherwise the sentence sounds too speculative.
- L276-277, two sentences: why cancer cells are more “vulnerable” to NTP-generated ROS? Please rephrase or add some information on that.
- L280: “production” à “accumulation”, see #24?
- L287-291: NTPAM injection was performed intratumorally only (L500), not systemically (e.g. intraperitoneally, to the chest cavity or bloodstream). It therefore could not be concluded whether “cytotoxicity in normal cells in a mouse model” was not observed. It is necessary to rephrase this part to emphasize that no side effects on normal tissue were detected under given experimental settings.
- L306: “induced “ à “mediated”, or “NTPAM treatment-related”, see #1.
- L307: please change “blocked” to e.g. “reduced”.
- L336: NTPAM effects on Hth7 and KTC2 are not presented in the manuscript or supplementary material, please add these data as supplementary.
- L370: please indicate how many replicates per data point were used.
- L378: “and” à “with”.
- L391 and L392: please expand “slpm” and “sccm”, respectively.
- L394-396: these two sentences do not flow well, please rephrase.
- L404-405: this sentence does not flow well with the next one, please rephrase.
- Subsection 4.11 from L475: in addition to #21, please add PCR conditions, amplicon length and indicate whether melting curve analysis was performed.
- Subsection 4.13: how many mice per treatment group were used?
- L516: please specify the temperature for antigen retrieval procedure.
Round 2
Reviewer 3 Report
The manuscript is markedly improved, most questions are satisfactory and constructively replied. The major concern is statistical methods, the authors may probably need to seek advice from a statistician. The following points need to be clarified.
- L40: please add “cells” after “cancer”.
- L48: please italicize EGR1.
- L51: please move “gene” to L52 after “(GADD45a)” and italicize GADD45a (two times on L52).
- L62: please add “major” after “five” (as there are other less prevalent types of thyroid malignancies).
- L84: EGR1 appear to regulate both TP53 (italicized and not “p53”) gene expression and TP53 (not italicized and not “p53”) protein function. Please correct this sentence.
- L86, 87: here, the gene names are meant. Those should be spelled correctly and italicized. Please bear in mind that gene names in rather old papers frequently do not correspond to current nomenclature.
- L89-90. Please provide appropriate gene names in a correct form.
- L.114: After the sentence ending with “Fig. 1C).”, please add the information appearing in the Discussion on L358-362 showing that NTPAM prepared by the described method displays stability and refer Supplementary Fig. 4 here first (perhaps, placing Supplementary Fig. 4 as Supplementary Fig. 1 would be more appropriate and logical).
- L118: please change “a time“ to “an activation time“.
- Fig. 1 legend: please add “of activation” after “120 min”.
- Fig. 1 legend and other places throughout the manuscript: Usage of the Student’s t-test is not appropriate for data in Fig. 1D (and many other places), and it is unclear what “groups” were compared. It would be expected to compare each point with the reference (0 min) for each cell line using a non-parametric one-way ANOVA followed by an appropriate post hoc test (e.g. Dunnet). Please redo statistical calculations where necessary throughout. In addition, the horizontal axis in Fig. 1D title sounds confusing, please rephrase, e.g. “time of medium activation with NTP (min)”. What are the error bars shown (SE, SD?) – please add this and whether the ratios were calculated using the propagation of uncertainty method or other to Methods/ Statistical analysis (as this relates also to some other figures in the manuscript).
- Fig. 2 legend: again, as in #11, Student’s test is suitable for normally distributed data. If the number of replicates in the experiments was rather small, impeding a meaningful test for normality, the non-parametric tests should be used. In Fig. 2CDE e.g. the Mann-Whitney U would be pertinent.
- L.167: After “Fig. 3F)”, please add the information appearing in the Discussion on L310-311, “because it showed the highest expression ratio and z score in BCPAP and 8505C cells” and mention that ATF4 and DDIT3 were also studied (Supplementary Fig. 3).
- L179-182, multivariable logistic regression. First, the model should be described e.g. in the footnote to Supplementary Table 2. Actually, it would be expected that each clinicopathological feature is separately tested as an outcome variable, while EGR1 expression (either in an ordinal or a continuous form in combination with other variable(s), e.g. Age) serves as an explanatory variable. Supplementary Table 2 suggests that another model was tested where the EGR1 low/high was an outcome. The results presented in Supplementary Table 2 also suggest that LN metastasis is positively associated with high EGR1 while Supplementary table 1 suggest the opposite. Furthermore, it is hard to believe that stepwise feature selection would result in a reduced model in which the variables with the p-values of 0.6-0.7 are present. All this may stem from an incorrect building of a model. Please recalculate in the way described above and report ORs instead of the regression coefficients.
- L182-184, the Kaplan-Meier analysis. I acknowledge that the analysis was performed on the whole TCGA data set, and statistical significance was not observed for the EGR1 high versus low groups. Considering the results from the previous version on the manuscript, it seems quite possible that subdividing the data set into more than 2 groups (e.g., using the tertiles or quartiles as cut off points instead of the median) and testing those may result in a statistically significant finding. The authors are strongly encouraged to perform such an analysis to figure out if this could be true (the KM analysis would need to be adjusted for multiple testing!). If a statistically significant difference is found, the Cox regression can be used with the appropriate group chosen as a reference.
- Fig. 3H. The graph does not look as a box plot, please correct. If the vertical axis is log-transformed, please add this to the axis title. If log2 TPM in the N and T groups are distributed nearly normally (please, check), that the Student’s t-test can be used safely here. In this case, please move the last sentence from Fig.3 legend to the 3H part.
- Fig. 3I: please add that the p-value is from the logrank test to the figure or figure legend.
- Fig. 4 ABEF: please use non-parametric test.
- Fig. 5 AB: correction for multiple testing is necessary. BEF – please use non-parametric test. B – post hoc test should be also applied. The last sentence in the legend is incomplete.
- Fig.6 AFGHI: please use non-parametric test. FG – post hoc test should be also applied.
- L269: please mention here the information on IP injection of NTPAM from the Discussion (L303-305) and Supplementary figure 2.
22.Fig. 7 EF: please use non-parametric test.
- L294: please change “be” to “in part be due to”.
- L354: please change “showing” to “characterized by”.
- L396-397: the sentence does not flow well, please rephrase. E.g. “To prepare NTPAM, RPMI 1640 medium was exposed to NTP for various periods of time which correspond to different NTPAM concentrations.”
- L399-400: Fig. 1 demonstrates that NTP was above the medium, not 1 cm from the bottom. Please rephrase.
- L401-402: please rephrase this sentence. E.g. “Cell treatment with NTPAM was performed by changing the culture medium for NTPAM.”
- L404: “triplicate” à “triplicates”.
- L422: “104” probably means 10 to the 4th power?
- L423: how many replicates were used?
- L508: “Paired nontumor tissues” probably means “Paired tumor and normal thyroid tissues”?
- L521, Transient transfection: please add more details on plate/well size, number of cells and replicates.
- L553: please add how long the antigen retrieval was performed.
- L561-565, Statistical analysis. This subsection should be extensively revised.
- Supplementary Fig. 3. Cell viability (in the legend) and cell proliferation (in the vertical axis title) are different terms. Please be consistent and use e.g. “viability” throughout. Please use nonparametric test with post hoc test.
- Supplementary Fig. 4. As suggested in #8, it is logical to make this figure Supplementary Fig. 1. Please also use nonparametric statistical test with post hoc test.
- Supplementary Table 2. Please see #14 and test other models.
